# Resection of Infrapatellar Fat Pad during Total Knee Arthroplasty Has No Impact on Postoperative Function, Pain and Sonographic Appearance of Patellar Tendon

**DOI:** 10.3390/jcm11247339

**Published:** 2022-12-10

**Authors:** Sławomir Michalak, Łukasz Łapaj, Arleta Witkowska-Łuczak, Paweł Chodór, Jan Zabrzyński, Jacek Kruczyński

**Affiliations:** 1Department of General Orthopedics, Orthopedic Oncology and Traumatology, Poznań University of Medical Sciences, 61-545 Poznan, Poland; 2Faculty of Medicine, Nicolaus Copernicus University in Torun, Collegium Medicum in Bydgoszcz, 85-067 Bydgoszcz, Poland

**Keywords:** infrapatellar fat pad resection, total knee arthroplasty, patellar tendon ultrasound

## Abstract

Routine resection of the infrapatellar fat pad (IFP) during total knee arthroplasty (TKA) is controversial, as it may result in shortening of the patellar tendon (PT) and anterior knee pain. This prospective study examined whether IFP excision during TKA affects joint function, anterior knee pain, PT dimensions and sonographic structure. A total of 65 consecutive patients undergoing TKA for osteoarthritis were randomized into two groups: IFP was resected in one and retained in the other. Patients were examined preoperatively, at 6 weeks and 6 months postoperatively: pain (Numerical Rating Scale—NRS), range of motion (ROM) and knee function (Knee Injury and Osteoarthritis Outcome Score—KOOS score) were evaluated; sonographic examination determined the length, structure and vascularity of the PTs. In both groups there were postoperative improvements in NRS and KOOS scores, although IFP resection did not influence clinical outcomes or sonographic parameters. At 6 weeks and 6 months postoperatively for both groups there were no differences between NRS scores (Mann–Whitney test, *p* = 0.511 and *p* = 0.579), ROM scores (Mann–Whitney test, *p* = 0.331, *p* = 0.180) or all KOOS subscores. IFP excision had no effect on sonographic parameters. This study suggests that IFP resection during TKA does not influence postoperative functional outcomes, pain scores, patellar tendon length and thickness, or sonographic structure.

## 1. Introduction

The infrapatellar (Hoffa’s) fat pad (IFP) is a structure formed from adipose tissue localized in the anterior compartment of the knee joint, localized extrasynovially with abundant innervation and vascularization [1,2]. Adipocytes in IFP are smaller than in storage fat and form structural, mechanically protective building fat [3]. There are numerous cell types apart from adipocytes to be found: macrophages, mesenchymal stem cells, mast cells, monocytes, and B and T cells [4]. Its physiological role is not fully understood and in arthritic knees the IFP is often enlarged with local synovial hypertrophy present [5]. There is evidence suggesting that it is an endocrine organ, capable of producing inflammatory cytokines involved in the pathogenesis of osteoarthritis (OA), while some authors indicate that it may actually protect the cartilage in arthritic joints [6,7]. Recent biomechanical study found that Hoffa’s fat pad becomes stiffer in OA due to fibrosis, causing distortion in force distribution throughout the knee joint [8]. Thus, with IFP’s secretory activity being shifted towards pro-inflammatory cytokines, Hoffa’s fat pad seems to have a substantial contribution to disease worsening [4,9,10]. During Total Knee Arthroplasty (TKA), complete or partial excision of the IFP is often performed to improve visualization and minimize the risk of interposition with the prosthesis [5,6,7].

Opponents of this technique raise the point that sacrificing the fat pad can lead to diminished blood supply to the patellar tendon (PT), which can potentially lead to its shortening [11,12,13]. This was observed in some clinical studies, although a number of papers showed no influence of IFP removal on postoperative length of the PT [14,15,16,17]. Another controversy associated with IFP excision is the incidence of anterior knee pain following TKA. Several studies demonstrated increased postoperative pain and lower functional scores in knees with resected IFP, yet other authors did not confirm that [6,18,19,20]. 

As data from the literature are inconsistent, we hypothesized that routine IFP resection during TKA does not compromise the postoperative improvement of knee function after six months. This prospective randomised study evaluated the effect of IFP removal on functional outcomes, incidence of pain, patellar tendon dimensions, sonographic echogenicity and vascularity.

## 2. Materials and Methods

### 2.1. Study Design

This prospective study included a series of 67 consecutive patients undergoing TKA because of idiopathic OA, with Knee Injury and Osteoarthritis Outcome Score (KOOS) pain domain chosen as an endpoint next to the Numerical Rating Scale (NRS) pain level. The number of patients was calculated a priori based on data from a pilot study of 10 patients, so that minimal clinically important differences between both groups could be verified for KOOS and NRS [21,22]. Using an online sample size calculator (ClinCalc) with the level of significance set at 0.05 and power of study set at 0.8, the minimal number of patients in each groups was calculated to be 31.

Inclusion criteria were the presence of severe unilateral idiopathic arthritis (Ahlback score II or more) and an informed consent from the patient. The following exclusion criteria were selected: secondary OA, previous surgical procedures within the affected knee, severe deformity (>15 deg. valgus >20 deg. varus), advanced OA in other joints, diabetes, advanced atherosclerosis of lower limbs. After inclusion, each patient was randomized using sealed envelope protocol [23]. Two groups were established: A—where total resection of the IFP was performed during TKA and B—where the IFP was preserved. Patients were blinded to the information if their fat pads were excised until the last follow-up examination was completed. All procedures were performed by two experienced surgeons (JK—24 cases (12 in group A and 12 in group B); PM—41 cases (24 in group A and 17 in group B)); depending on their intraoperative decision, either cruciate-retaining (CR) or posterior cruciate substituting (PS) cemented Stryker Triathlon implants were used. As two patients were lost to follow-up, ultimately the cohort consisted of 65 patients.

### 2.2. Surgical Procedures

All procedures were performed using anteromedial approach; torniquet was applied in each case. In patients from group A, the IFP was bluntly divided from the patellar ligament and resected using electrocautery. In patients from group B, the entire IFP was retained, although surgeons incised it at the medial border of the patellar ligament to gain exposure of the knee joint. In all cases, the anteromedial joint capsule was routinely released from the tibia, however, any fragments of IFP attached to it were not disturbed. The surgeries were performed according to the concept of mechanical alignment; femoral components were implanted using posterior referencing technique, while the rotation of tibial components was established parallel to a line drawn from the posterior cruciate ligament to the medial third of tibial tuberosity. In all cases the patella was neither resurfaced nor denervated, although large patellar osteophytes were removed if present.

### 2.3. Clinical and Sonographic Evaluation

Clinical and sonographic evaluation of the knee joints was performed preoperatively, at 6 weeks and 6 months follow-up. The study was approved by the Institutional Bioethical Committee; all participants gave informed consent.

Clinical examination included evaluation of the range of motion (ROM) and presence of local pain. The intensity of pain was evaluated using the (NRS) as this scale is very easy to understand by most participants. Patients were asked to choose a fixed numeric value between 0 and 10 which best fit their local pain sensation (with 0 corresponding to no pain and 10 corresponding to the worst pain imaginable). Subjective joint function was examined using the (KOOS) during each check-up. For each KOOS subscale—pain, other symptoms, function/activities of daily living (ADL), function in sport and recreation (FSR), and knee-related quality of life (QOL)—the results were transformed to the 0–100 scale, where 0 corresponds to the worst function and 100 indicates a well-functioning knee.

Sonographic evaluation and measurements were performed by an experienced musculoskeletal radiologist (A.W.-L.) using a 5–12 MHz linear probe (Philips Affiniti 50G, Philips, Amsterdam, The Netherlands). The protocol included measurements of PT length (at the posterior border) and thickness (1 cm below the distal tip of patella). The sonographic echogenicity of PT was classified as normal, increased or decreased. In order to minimize the anisotropic effect, the examination was always performed with the probe perpendicular to the long axis of PT with the knee placed on a bolster to ensure identical knee flexion in each case. Color Doppler (CD) mode was used to determine the vasculature of the tendon based on the semiquantitative Gisslén scale [24]. In this system, the following scores can be applied: 0—no vasculature, 1–increased vascular supply outside the ligament, 2—one or two vessels present within the ligament, 3—more than three vessels present; scores 2 and 3 were classified as increased vascularity of the tendon. Additionally, the total length of all vessels (TLV) was measured, as described by Cook et al. [25]. Briefly, in CD mode, the probe was placed longitudinally in the central part of the ligament and each vessel with a diameter of 1 mm or more was measured (meandering vessels were first divided into max. three smaller sections). Subsequently, the length of all vessels and vessel sections was summed up.

### 2.4. Statistical Analysis

The data was analyzed with the use of GraphPad Prism v.8.0.1 (GraphPad Software, La Jolla, CA, USA) and Statistica 13.3 (TIBCO Software, Palo Alto, Santa Clara, CA, USA) software packages. In all cases *p* < 0.05 was chosen as a threshold for significance and variables were tested for normality using the Shapiro–Wilk test. The Paired *t*-test, Mann–Whitney U and Wilcoxon tests were applied to compare the data; the Friedman, ANOVA and post hoc Dunn tests were applied for multivariate analysis. The Fisher exact test and the Chi-squared test were employed to compare descriptive characteristics. Statistical strength of parametric and non-parametric tests was also evaluated. 

## 3. Results

### 3.1. Cohort Characterization

Both groups of patients had comparable demographic characteristics (Table 1). There were no statistically significant differences between their age and BMI, the progression of OA was also similar. Two patients were lost to follow-up as they did not want to undertake long distance travel to our institution; when interviewed on the phone at 6 months, both were satisfied with the outcome of their surgeries and had NRS scores of 2 and 3.

### 3.2. Knee Injury and Osteoarthritis Outcome Score Evaluation

In both groups of patients, during post-op evaluations, scores for pain, symptoms and function related to daily life activities improved significantly; less marked improvements were seen for sports and recreation-related function as well as knee-related quality of life (Figure 1). 

Statistical analysis using the Mann–Whitney U test demonstrated that KOOS scores did not differ preoperatively, and IFP removal did not affect these scores, as there were no statistically significant differences between values of all KOOS subscales (Table 2).

### 3.3. Pain Intensity, Location and ROM

Most patients complained of preoperative pain within the medial compartment and/or the lateral part of the knee joint, with no cases of anterior knee pain; following TKA, an improvement was observed in both groups (Table 3). At 6 months in most cases pain localized on the medial side was localized near the tibial component; there were no radiographic signs of overhang, implant malalignment, loosening or infection. IFP resection did not influence incidence of pain in all compartments, and such pain was not observed after 6 months postoperatively (Table 3).

Postoperative NRS pain scores at 6 weeks and 6 months were significantly lower (Figure 2) than the preoperative values in both groups (Dunn test, *p* < 0.0001 in all cases). The resection of the IFP did not affect the NRS scores, as there were no statistically significant differences between both groups at 6 weeks and 6 months (Mann–Whitney U test, *p* = 0.511 and *p* = 0.579, respectively). 

ROM values initially decreased in both groups postoperatively (Figure 2), although the change was statistically significant only in group B (paired *t*-test, *p* = 0.0846; Wilcoxon test, t *p* < 0.0001 for groups A and B, respectively). This was followed by a statistically significant improvement in both groups (paired *t*-test, *p* = 0.0132; Wilcoxon test, *p* = 0.001 for groups A and B, respectively). IFP excision did not influence postoperative ROM and there were no differences between both groups at 6 weeks and 6 months (Mann–Whitney U test, *p* = 0.331, *p* = 0.180, respectively).

### 3.4. Sonographic Evaluation

Performing TKA did not have an effect on the length but was associated with a significant increase PT thickness postoperatively in both groups (both *p* < 0.0001; Dunn test); this was followed by a minor decrease in thickness at 6 months (Figure 3). IFP resection did not affect the PT length and thickness as there were no differences between both groups preoperatively, at 6 weeks and 6 months (Table 4).

Most knees included in this study had normal PT with echogenicity preoperatively, however at 6 weeks post-op, an increase of the number of hypoechogenic tendons was observed, which then subsided at 6 months post-op (Table 5). Statistical analysis did not demonstrate any difference between the preoperative echogenicity of PTs in both groups (Fisher’s exact test, *p* = 0.7084); IFP resection also did not affect this parameter at 6 weeks and 6 months (Fisher’s exact test, *p* = 0.6975, *p* = 0.1478, respectively).

Preoperatively in both groups no vessels were detected in PTs (Table 5). Postoperatively vascularity was increased at 6 weeks and to a lesser degree at 6 months; in most cases Gisslén scores 2 and 3 were found. The Mann–Whitney test demonstrated that FP resection did not influence the vascular supply as there were no differences between both groups at 6 weeks and 6 months for Gisslén score and TLV.

### 3.5. Statistical Power Analysis

As the sample size in this study is limited, an analysis of power of statistical tests was performed. For non-parametrical tests comparing results between both groups, the power in all cases was 0.99. In group A for parametric tests comparing data at different points, the power was in the range 0.42–0.99, while in group B, the power was 0.99 in all cases.

As there was a small number of patients with PS implants in group A, additional statistical evaluation was performed with these cases excluded. This transformation did not have a significant influence on results of statistical tests comparing both groups, yet caused minor changes in outcomes of statistical tests regarding KOOS scores in group A. These patients were included in the study nevertheless, as this impacted the power of statical analysis.

## 4. Discussion

This prospective randomized study demonstrated that IFP excision during TKA had no effect on postoperative ROM, pain levels, function of the knee joint or sonographic parameters of PTs. Routine excision of the IFP during TKA is a subject of debate, and a survey performed in 2004 by the National Joint Registry of England and Wales demonstrated that 27% of surgeons performed a total, and 59% a subtotal, excision of the IFP, while 14% retained it [26]. The rationale for resecting IFP partially or totally is to gain enhanced surgical exposure of the lateral compartment [5,6,7,15]. However, there are concerns regarding the influence of IFP excision on the postoperative vascular supply of the patella and patellar tendon. Cadaveric studies demonstrated that IFP resection may compromise vascular supply to the tip of patella and PT [27,28,29]. Still, data from clinical studies is conflicting and there is no clear evidence supporting either resecting or sparing the IFP. 

Several studies demonstrated that IFP excision is associated with increased anterior knee pain: Macule et al. reported increased pain 6 months following TKA, while in a study by Nisar et al. elevated pain was observed at 1–2 months post-op [14,30]. Pinsornask et al. demonstrated comparable pain intensity at 2 months, but increased pain in knees with IFP resection at 12 months; other authors also indicated anterior knee pain associated with Hoffa fat pad resection at follow-ups longer than 6 months [7,20,31]. In the present study, NRS pain scores were comparable in both groups, thus suggesting that IFP resection is not associated with anterior knee pain. While we cannot exclude that similar differences could be observed in our patients at longer follow-up, it should nevertheless be underlined that anterior knee pain following TKA is affected by many factors, such as patellar resurfacing or implant alignment, and there is no evidence indicating a pivotal role of IFP resection [32]. 

Some authors have pointed out that IFP excision compromises vascular supply to PT and can potentially lead to the formation of a fibrous scar and shortening of the tendon as well as decreased flexion [5,7,13,16,33,34,35]. Some authors demonstrated shortening of PT typically seen at 6–12 months, yet without decreased flexion [5,30]. This is consistent with findings from the present study, as no decrease of PT length and ROM was observed in knees with IFP resection, although a decrease of ROM was seen in both groups postoperatively, similar to the findings reported by other authors [36,37,38,39]. 

In this study, KOOS scores improved at 6 weeks and 6 months and were not affected by IFP resection; similar results were reported in other papers [18,40]. Some studies demonstrated limited increases in functional scores between 6 and 12 months post-op; similar improvements could be expected in patients from this series [40,41].

Sonographic examination of knees from this study demonstrated increased postoperative thickness and vascularity of PTs as well as changes in their echogenicity. Gaulrapp et al. demonstrated PT thickening at 4 weeks post-op and concluded that postoperative sonographic appearance of PT is similar to that seen in tendinopathy [42]. Consequently, it could be expected that sonographic alterations seen in PTs following TKA would be associated with clinical symptoms, predominantly pain; still, after 6 months none of the patients reported anterior knee pain. 

The authors acknowledge certain limitations of this study. First, the synovial fluid was not examined preoperatively. As the IFP and synovial membrane are involved in the inflammatory process, it is possible that cytokine levels varied among patients potentially affecting pre- and postoperative pain. Similarly, some aspects of the surgical technique (referencing, mechanical alignment) as well as the fact that patellar denervation was not performed could potentially affect the incidence of postoperative pain [43,44].

As in other papers, the number of patients was relatively low but the statistical power of this study was chosen to allow a meaningful comparison between both groups. This was done based on data on minimal clinically important differences obtained from the literature, however their values differ among publications [21,22]. Longer follow-up could potentially improve the quality of the results, however most studies demonstrated that clinically relevant improvements are observed within the first 3–6 months post-op [40]. All patients received the same type of implant with single radius geometry, no patellar resurfacing was performed and PS knees were used in six cases; these factors could affect the incidence of postoperative pain. Some authors demonstrated low incidence of postoperative pain for implants with single radius design, while other suggested the opposite for fixed bearing PS knees [45,46]. The relationship between patellar resurfacing and osteophyte removal and pain following TKA is also controversial [18]. In most patients from this study, the etiology of postoperative pain could not be determined based on clinical and radiographic studies; similar findings were also reported by other authors [47].

Another weakness of the present study is related to the subjective nature of the sonographic evaluation, which predominantly applies to echogenicity. This could potentially be eliminated by the use of MRI, however, as the knee would have to be flexed for consistent measurements. This is problematic in postoperative examinations, as the tendon would be in close proximity to the femoral component. In such alignment, suboptimal image quality could be a serious risk even despite the use of metal artifact reduction sequences.

Although methods allowing for quantitative evaluation of echogenicity based on greyscale data from calibrated images are available, they are still prone to operator error—predominantly inconsistent probe tilting. It is for this reason that a semiquantitative method was chosen in the present study, and in order to minimize this effect, all studies were performed by one musculoskeletal radiologist, and careful knee positioning was performed prior to each examination. 

## 5. Conclusions

This study suggests that IFP resection during TKA does not influence postoperative functional outcomes, anterior knee pain, or patellar tendon length and thickness. Postoperative changes in PT vasculature and echogenicity vary between individuals and do not seem to be associated with anterior knee pain. Still, further studies are needed to precisely verify any potential relationship between sonographic and clinical findings following TKA.

## Figures and Tables

**Figure 1 jcm-11-07339-f001:**
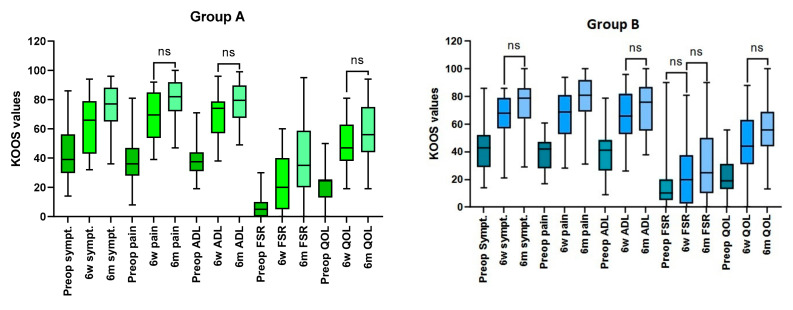
**Group A—IFP resected, Group B—IFP preserved.** KOOS scores in both groups (preoperative, 6 weeks and 6 months) of patients obtained at all check-ups. In comparison to preoperative values, all scores significantly improved at 6 months; when scores from consecutive examinations were compared as statistically significant (Friedman and post hoc Dunn tests) improvement was observed for most of them (with few exceptions, denoted with the ns markings) The following abbreviations for KOOS domains were used: sympt.-symptoms, ADL—activities of daily living, FSR—function–sports and recreation, QOL—quality of life.

**Figure 2 jcm-11-07339-f002:**
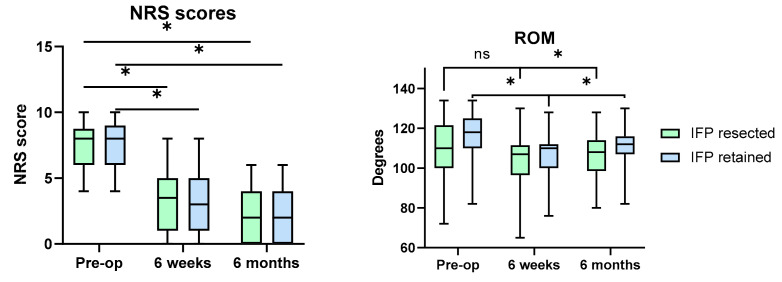
NRS scores and ROM in patients with resected (green) and retained (blue) IFP preoperatively, at 6 weeks and 6 months follow-up. Asterisks indicate statistically significant difference between values; ns—lack of thereof.

**Figure 3 jcm-11-07339-f003:**
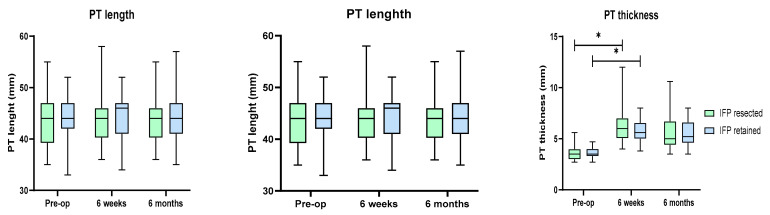
PT length and thickness in both groups of patients preoperatively and at both postoperative examinations. Asterisks denote significant differences between values.

**Table 1 jcm-11-07339-t001:** Cohort characteristics. BMI–body mass index.

Parameter	Group A	Group B
Number of cases (M/F)	32 (27/5)	33 (26/7)
Mean age (range)	68.3 (53–78)	68.1 (56–81)
Mean BMI (range)	29.7 (25.1–34.8)	30.1 (26.2–34.8)
Ahlback scores (II/III/IV/V)	(12/19/1/0)	(10/21/2/0)
TKR types—CR/PS	26/6	33/0
Varus/Valgus knees	28/4	29/4

**Table 2 jcm-11-07339-t002:** Comparison of KOOS scores between both groups at 6 weeks and 6 months; *p* values of Mann–Whitney U tests are presented; the following abbreviations for KOOS domains were used: ADL—activities of daily living, FSR—function–sports and recreation, QOL—quality of life.

Evaluation Point	Symptoms	Pain	ADL	FSR	QOL
pre-op	*p* = 0.9790	*p* = 0.6592	*p* = 0.4421	*p* = 0.0833	*p* = 0.9364
6 weeks	*p* = 0.6123	*p* = 0.4503	*p* = 0.8017	*p* = 0.8010	*p* = 0.6869
6 months	*p* = 0.7022	*p* = 0.6543	*p* = 0.2620	*p* = 0.3610	*p* = 0.9660

**Table 3 jcm-11-07339-t003:** Pre- and post-operative incidence of severe pain presence of severely (NRS at least 4) in compartments of examined knee joints. The data regarding painful compartments is given in the following order: medial/lateral/anterior; Number of patients who developed pain in previously asymptomatic compartments was given in brackets. Comparison of both groups presents results of Chi-squared test except for one case marked with asterisk, where Fisher’s exact test was used to verify potential differences for a small group; n.a.—not applicable as there was no anterior knee pain.

Knee Pain Evaluation Point	Pain in Compartments Group A—med./lat./ant.	Pain in Compartments Group B—med./lat./ant.	Comparison of Group A and B
Preoperative	29/6/0	29/2/0	*p* = 0.4136/*p* = 0.1195/n.a.
6 weeks post-op	6/5/0	9/8/3	*p* = 0.2386/*p* = 0.5767/*p* = 0.2385 *
6 months post-op	6(1)/1(1)/0	8/2(2)/0	*p* = 0.8649/*p* = 0.9782/n.a.

**Table 4 jcm-11-07339-t004:** Comparison of PT length and PT thickness between groups A and B; *p* values of Mann–Whitney U test are presented.

Evaluation Point	PT Length	PT Thickness
Pre-op	*p* = 0.4027	*p* = 0.4529
6 weeks	*p* = 0.1266	*p* = 0.5238
6 months	*p* = 0.4180	*p* = 0.6174

**Table 5 jcm-11-07339-t005:** Echogenicity and vasculature of PTs. The echogenicity is presented as normal/decreased/increased; Median TLV values are presented with min-max values in brackets.

Parameters	Group A	Group B
PT echogenicity–pre-op TLV—pre-op	28/3/1 0 (0)	28/5/0 0 (0)
PT echogenicity–6 weeks TLV—6 weeks	19/12/1 9 (0–32)	22/11/0 7 (0–36)
PT echogenicity–6 months TLV—6 months	24/6/2 0 (0–20)	30/3/0 0 (0–18)

## Data Availability

Data is available in data repository https://doi.org/10.6084/m9.figshare.21507057.

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
