# Peer review of "Resection of Infrapatellar Fat Pad during Total Knee Arthroplasty Has No Impact on Postoperative Function, Pain and Sonographic Appearance of Patellar Tendon"

_jcm, 2022, doi:10.3390/jcm11247339_

Round 1

Reviewer 1 Report

Row 33 - Total Knee Arthroplasty - it is not required to use uppercase letters here

It is also important to state if patellar denervation was performed in these cases or not (with electrocauterization)

Is there a difference between Visual Analogue Scale and Numerical Rating Scale? The first one is more used as a terminology in the literature. Please explain if there are any variances.

In the orthopaedic world, we know what KOOS means. This is not a self-reliant thing for multidisciplinary journals like JCM. Please explain the abbreviation of KOOS in the dedicated section. I would also suggest changing the name of the section in Knee Injury and Osteoarthritis Outcome Score (instead of using the abbreviation)

Figure 1, Table 2. QOL, ADL, FSR etc. abbreviations should be defined in the legend of the figure/table.

For Table 2 - is there a reason for not having it compared to the preoperative scores? You only reported 6 weeks and 6 months.

You mentioned 67 patients. If I look carefully at Table 1, and add the cases, they add up to 65 patients. What is the reason for the exclusion in two cases? In Table 5, the cases do not add up to 67 again.

Figure 3. Replace Lenghth with "Length"

About 50% of your references are 10 years old or more. Can you please address this?

Author Response

We would like to thank the editor and reviewers for their time and effort as well as all critical remarks. We did our best to improve the paper and implemented nearly all of the suggestions :

  1. Line 33 Total Knee Arthroplasty – the capitals were corrected

  2. Patellar denervation - this is a valid point – it was not performed in any of the patients – and the missing information was inserted into the materials & methods section as well as limitations

  3. VAS vs NRS scores - in contrast to VAS in the NRS scoring system the patient chooses one “fixed” value 0,1,2…10. This is briefly explained in line 107-109 – we changed the word “number” to “fixed numeric value” for clarity; the scale was chosen as it was easier to apply to the cohort

  4. KOOS abbreviation – we understand that this may not be clear to all readers; the KOOS was explained the first time it appears In the text – line 17 we also explained the name in the abstract and as suggested changed the caption of section 3.1

  5. Figure 1 and table 2 had no explanations of abbreviations – the explanations were added

  6. Missing preop KOOS scores in table 2 – pre-op values did not differ between both groups so we skipped this initially; this is however a valid point and the data was added into table 2

  7. Lack of clarity - 65-67 cases – thank you for this point – as demonstrated in table 1 there were 65 cases in total as 2 patients were lost to follow-up. This was clarified in the text and corrected in the abstract; We also clarified why patients were lost to follow-up (long distance travel – yet they were satisfied with the surgery)

  8. Figure 3 spelling mistake - the spelling was corrected

  9. the references are 10+ years – that is a valid point - we reviewed the literature and included newer papers; we would still like to point out that newer literature was generally in line with the older publications

Reviewer 2 Report

Dear authors,

thank you very much for giving me the opportunity to review this work. As you point out it is not known whether the resection of the infrapatellar fat pad (IFP) has an effect on the patient's outcome. Therefore, as IFP resection is often routinely performed, the topic of your work is of high clinical interest.

In general, this work is well structured and well written. The results are clearly presented and support your conclusion.

However, i would like to raise some methodological issues:

- Since it was an prospective randomized study i expect a definition of the primary outcome variable and a calculation of the sample size before the study takes place Was that performed? If not, why did you investigate this number of patients?

- Were the patients blinded to randomization? If not, at what time point did they know if IFP resection was performed/planned?

- The surgeries were performed by two experienced surgeons. How was the distribution of the surgeons among the two groups?

Best regards

Author Response

We would like to thank the editor and reviewers for their time and effort as well as all critical remarks. We did our best to improve the paper and implemented nearly all of the suggestions :

  1. Endpoint information was missing – thank you for raising this point - KOOS pain domain and NRS were chosen as the primary endpoint, as was done in other papers comparing TKA techniques

https://doi.org/10.1007/s00402-017-2797-5

Sample size was calculated to allow for determination of minimally clinically important differences for NRS and KOOS using the CliniCalc website as was done by other authors. We chose minimal clinically significant values for both parameters (NRS =2; KOOS pain =10) based on data from the literature (the references were also provided); Based on initial data from a pilot study of 10 patients we had the necessary means/SDs to calculate the sample size

https://doi.org/10.1016/j.arth.2021.02.038

https://doi.org/10.3390/jcm9030837

  1. We did not clarify how patients were blinded – this is a valid point as this could affect the results - the patients were blinded till their last follow-up visit; this was updated in the text – line 78-79.

  2. There was no data on surgeon involvement in each group – this is a valid point as the technique could affect the overall outcome - the of procedures distribution was updated in the text – line 80-81

Reviewer 3 Report

Manuscript: “Resection of infrapatellar fat pad during total knee arthroplasty has no impact on postoperative function, pain and sonographic appearance of patellar tendon.”

The introduction is very short. It should be improved.

A better background on IFP and OA should be reported. In particular, it is clear that IFP participates in OA pathology (OA is a whole joint disease). It is reported to be an inflamed and fibrotic tissue in OA patients. Moreover, this tissue has been recently tested from the mechanical point of view demonstrating as this tissue is stiffer compared to other adipose tissues likely due to the fibrotic changes occurring during the pathology.

Section 2: inclusion criteria are missing.

Line 53: how did the authors perform the randomization?

Demographic data of the enrolled patients should be reported at the beginning of the results. Statistical analysis should be applied to understand if there are differences between the two groups of enrolled patients. Body mass index must be added and compared.  Kellgren-Lawrence of the patients should be added.

Surgical procedure should be briefly described in the methods.

Lines 93-100: this part should be separated and titled” statistical analysis”.

Lines 99-100:  how did the authors check the normality of the data?

Figure 1 is unclear.  The author should highlight the statistically significant comparisons and not the data where no statistical significance was showed.

 Table 3 is unclear. Here, the authors reported the pain evaluated in medial/lateral/anterior compartment. Did the authors evaluate pain with NRS scale (range 0-10) or KOOS pain?  Did the authors report the mean/median of the score for each compartment? Why did the authors use Fisher test in one case? The authors reported “new origins of pain included in brackets”. This point needs clarification. What does it mean that patient of group A had 6 (1) in the 6 months follow-up?

Did the authors evaluate synovial inflammation of these patients? This is because, first infrapatellar fat pad and synovial membrane seem to function as an anatomo-functional unit, and second, because it has been demonstrated that synovitis is linked to OA pain and thus, it might influence the results.

As the authors reported in the limitations, the sonographic evaluation is subjective. It would be more interesting to evaluate patellar ligament parameters and synovitis on MRI images. This point should be added in the discussion.

Author Response

We would like to thank the editor and reviewers for their time and effort as well as all critical remarks. We did our best to improve the paper and implemented nearly all of the suggestions :

  1. The introduction was indeed short – we recognize the important biomechanical and biochemical role of the IFP – synovium complex; we consequently improved the introduction and the part focusing on the role of IFP was added

  2. Inclusion criteria are missing – we clarified their description, details were provided in the text – line 71-72

  3. Line 75 randomization – the patients were asked to draw closed envelopes with numbers as described by Doig et al. Relevant literature citation describing the protocol was added.

  4. Table 1 should be moved to the results + data was missing – this was done ; the data on BMI was provided; since we used Ahlback scoring system for OA progression evaluation – we provided these scores. There were no differences between these parameters in both groups – this was stated in this section as well.

  1. Surgical procedure should be briefly described – we included the critical points – alignment, referencing technique etc.; also the alignment / referencing was discussed in limitations

  1. Lines 133- 142 – the caption Statistical analysis was added; we also expanded the part regarding group size as requested by reviewer 2 however as this was critical for sample size calculation that part was described in study design

  1. Lines 99-100 – Saphiro-Wilks test was used to check normality of data; this was described in line 137

  1. Fig. 1 We do recognize the fact that significant values should be presented in the figures and insignificant differences would consequently be omitted. Still in this particular case as there were differences between preop-6w; 6w-6m and preop-6m the figure would be cluttered and difficult to understand ( two lines of markers on top of each other). Consequently we retained this “somewhat unorthodox” notation as it serves the purpose of clarity and easy understanding of the figure however we expanded the description of the figure to clearly state w

  1. Table 3 is unclear – This table conveys the information regarding painful knee compartments – this was done to verify if there is a difference in severe anterior knee pain between both groups; We clarified the description – provided information on the minimal NRS score which qualified each case included in this table (NRS>=4); we also clarified that the new origins of pain were compartments which were reported as painful while being asymptomatic previously; As this data was analysed using contingency tables we examined it with Chi2 tests. The only exception was one position where due to a small group size and “intuitive” difference 0 vs 3 the Chi2 test was verified with Fischers test – we were advised to do so by a biostatistician.

  2. Synovial fluid inflammation – this was not performed, as the reviewer indicated a valid point this was included in the limitations lines 289-292

  3. MRI – this would work well for the pre-op examinations, however as there were concerns regarding metal artifacts even in MARS sequences and ultrasound was used in most studies this method was chosen even despite metal artifact reduction sequences. Still, we agree that MRI provides valuable data and included this in the limitations section.

Pilania K, Jankharia B. Magnetic Resonance Imaging of Complications in Total Knee Arthroplasty: A Pictorial Essay. Indian J Musculoskelet Radiol 2019;1(1):21-26. DOI: 10.1016/j.arth.2020.05.052

Sato, Y., Kösters, A., Rieder, F., Sasho, T., Müller, E., & Wiesinger, H.-P. (2020). Quantitative Analysis of Patellar Tendon After Total Knee Arthroplasty Using Echo Intensity: A Nonrandomized Controlled Trial of Alpine Skiing. The Journal of Arthroplasty, 35(10), 2858–2864. https://doi.org/10.1016/j.arth.2020.05.052 

Round 2

Reviewer 3 Report

No additional comments.